# Intestinal Microbiota Interventions to Enhance Athletic Performance—A Review

**DOI:** 10.3390/ijms251810076

**Published:** 2024-09-19

**Authors:** Bharati Kadamb Patel, Kadamb Haribhai Patel, Chuen Neng Lee, Shabbir Moochhala

**Affiliations:** 1Department of Surgery, Yong Loo Lin School of Medicine, Level 8, NUHS Tower Block, Singapore 119278, Singapore; surbkp@nus.edu.sg (B.K.P.); surlcn@nus.edu.sg (C.N.L.); 2Temasek Polytechnic, School of Applied Sciences, 21 Tampines Ave 1, Singapore 529757, Singapore; kadamb@tp.edu.sg; 3Department of Pharmacology, Yong Loo Lin School of Medicine, Block MD3, 16 Medical Drive, Singapore 117600, Singapore

**Keywords:** gut microbiota, endurance, physical performance, probiotics, athletes, exercise, performance enhancement

## Abstract

Recent years have witnessed an uptick in research highlighting the gut microbiota’s role as a primary determinant of athletes’ health, which has piqued interest in the hypothesis that it correlates with athletes’ physical performance. Athletes’ physical performances could be impacted by the metabolic activity of the assortment of microbes found in their gut. Intestinal microbiota impacts multiple facets of an athlete’s physiology, including immune response, gut membrane integrity, macro- and micronutrient absorption, muscle endurance, and the gut–brain axis. Several physiological variables govern the gut microbiota; hence, an intricately tailored and complex framework must be implemented to comprehend the performance–microbiota interaction. Emerging evidence underscores the intricate relationship between the gut microbiome and physical fitness, revealing that athletes who engage in regular physical activity exhibit a richer diversity of gut microbes, particularly within the Firmicutes phylum, e.g., *Ruminococcaceae* genera, compared to their sedentary counterparts. In elite sport, it is challenging to implement an unconventional strategy whilst simultaneously aiding an athlete to accomplish feasible, balanced development. This review compiles the research on the effects of gut microbiota modulation on performance in sports and illustrates how different supplementation strategies for gut microbiota have the ability to improve athletic performance by enhancing physical capacities. In addition to promoting athletes’ overall health, this study evaluates the existing literature in an effort to shed light on how interventions involving the gut microbiota can dramatically improve performance on the field. The findings should inform both theoretical and practical developments in the fields of sports nutrition and training.

## 1. Introduction

Research on the intricate realm of our gut microbiota is expanding, recognizing it as an enigmatic and concealed organ [1]. Thanks to the latest developments in culture-based methodologies and emerging genomic approaches, it is now conceivable to gain insight into how the microbiome influences human health and disease (culture-omics) through its activities [2]. Exercising vigorously has long been believed to be an efficient means to both prevent and hasten the onset of a number of ailments [3]. There has been considerable concern about a possible connection between gut microbes and physical exertion, namely intense competitive sports. This correlation could perhaps offer a rationale for the positive effects of physical activity on the general well-being of individuals. The performance of athletes may experience a hit when their gut microbial composition undergoes adverse changes due to insufficient endurance training, excessive or inconsistent physical activity, or both [4]. Addressing the fundamental physiological components that foster the outstanding performance of elite athletes in endurance exercises has been a long-standing objective of athletics research and continues to be an essential field of emphasis [5]. In recent years, the association between athletes’ athletic abilities and their intestinal microbes has gained significant weight. This is partly because empirical evidence has revealed the crucial effect of the gut microbiota on athletes’ overall wellness [6]. Additionally, there is some evidence implying that the metabolic activity of the microbial community prevalent in the athletes’ large intestine can influence their performance in sports. The microbiota of an individual signifies an assortment of microbial organisms which live in a mutually beneficial connection with the human body. It is estimated that it consists of around 10^14^–10^15^ bacteria [7]. Complex assemblages of microorganisms that reside in different settings have become crucial participants in maintaining ecological equilibrium and human well-being. These varied collections of bacteria, viruses, fungi, and other microbes flourish in environments that span from the deep oceans to the human gastrointestinal tract [8]. According to estimates, the human gastrointestinal tract has a wide variety of bacterial species, with more than 1000 in total. Firmicutes, Bacteroidetes, Actinobacteria, Proteobacteria, and Fusobacteria are the primary phyla of intestinal microbes that we have identified. Moreover, the human gastrointestinal tract comprises approximately 2000 bacterial species that have not been cultured [9]. Firmicutes and Bacteroidetes collectively account for approximately 90 percent of the gut microbiota. The gut microbiota demonstrates stability, resilience, and a mutually beneficial relationship with the host in a state of good health. An extensive study was conducted to determine the precise characteristics of a “healthy” gut microbiota and its correlation with the physiological functions of the host [10].

The intestinal microbiota plays a crucial role in various metabolic processes, including the fermentation of raw carbohydrates to facilitate their breakdown and absorption, as well as in the acquisition and preservation of energy. This specific attribute has probably exerted significant evolutionary pressure, resulting in bacteria emerging as symbiotic creatures within the human body [11]. Moreover, it has a significant impact on mental well-being in humans through the gut–brain axis, a complex communication network. This effect extends beyond the microbiota’s primary activities [12]. It is critical to highlight that the gut microbiota engages in complex interactions with the immune system, thereby facilitating the transmission of signals that enhance immune cell maturation and the establishment of proper immunological functions under optimal conditions [13]. The gut microbiota displays significant variability in its makeup. Factors such as age, ethnicity, lifestyle, and dietary choices influence the variation in the physiological diversity of the healthy gut microbiota [14]. 

The influence of the gut microbiota on athletic performance and endurance is a growing area of investigation within numerous aspects of human health. Recent studies indicate that the makeup and operation of gut bacteria can have a substantial impact on an athlete’s stamina, recuperation, and overall athletic performance [15]. An imbalance in the gut microbiota‘s composition, commonly known as dysbiosis, can link to the onset or progression of various diseases [16]. Assessing the possible beneficial or detrimental effect of a move is often difficult. In the field of microbiome studies, recent research has investigated the use of innovative treatment methods, including probiotics, synbiotics, postbiotics, and prebiotics. These approaches target disorders associated with dysbiosis, with the goal of alleviating disease symptoms [17,18]. This review intends to investigate the possible connections between host microbiota and physical performance in athletes to improve overall health and endurance performance.

The microbiome of the gastrointestinal tract has been the focus of a plethora of intriguing scientific investigations across the past ten years. Recent developments in next-gen sequencing and metagenomics have culminated in substantial consideration being given to the relationship between exercise and the microbiota. Additionally, certain studies have provided evidence of a possible two-way connection between physical exercise and the gut flora, suggesting the feasibility of controlling this interaction. Therefore, this has the potential to provide a practical way to influence athletic performance by manipulating the microbiome, which presents a significant opportunity for many top athletes and their teams [19]. Supplementing the athlete’s microbiota with various strategies, such as microbe-friendly bacteria, e.g., probiotics, prebiotics, synbiotics, and postbiotics, improves the gut microbiota’s efficiency. This allows it to modulate the host’s energy metabolism, influence immunological responses, govern oxidative stress levels, and impact other aspects. A thorough investigation into several types of microbiota-targeted therapies reveals their enormous potential for advancement in the sports industry as possible aids for athletes engaged in rigorous training. They have the ability to accumulate and refill a greater amount of energy for the body throughout training or competition, akin to the way marathon runners replenish electrolytes and water during races [20]. 

A growing body of clinical evidence suggests that physical activity influences the overall makeup of the intestinal microbiota; more specifically, athletes’ microbial populations may be more diverse and enriched with species that promote physical well-being, in contrast to those of sedentary individuals [21]. The association involving the composition and diversity of microorganisms in the digestive system and the immune system, energy metabolism, nutrient absorption, and oxidative capacity of athletes during various forms of exercise has already been established [22]. These attributes are tightly linked to athletic performance. Therefore, the discovery of compounds that are capable of exerting an impact on the gut microbiota has emerged as a promising approach to enhance athletic performance in athletes.

## 2. Materials and Methods

A literature review was performed. The results have been obtained by direct online access to the following databases: PubMed, Web of Science (WOS), Scopus, Google Scholar, and Embase.

We examined articles published in any country, by any organization or individual researcher, authored in English. These articles were required to be available in full text via Open Access. The search was confined to items published between 2005 and 2024.

The employed search tactics were “endurance training” AND “probiotics”; “athletic performance” AND “probiotics”; “endurance training” OR “athletic performance” AND “probiotics”; “endurance training” OR “athletic performance” AND “probiotics,” prebiotics, synbiotics, postbiotics, and clinical investigations on sports endurance employing gut microbiota treatments.

The search was not limited by exercise type, intensity, gender, clinical condition, sample size, species, publication year, publication status, or follow-up duration. 

## 3. The Key Roles of the Gut Microbiota in Boosting the Overall Health and Performance of Athletes

The interaction involving the microbes in the gut and exercise is dual in origin. Physical activity, particularly endurance exercise, can modify the overall makeup of the microbiota as a result of elevated oxidative stress [21], higher intestinal permeability, an imbalance of electrolytes, and diminished glycogen stores, among other factors. The microbiota of the gut plays a crucial role in improving the integrity of the intestinal wall and promotes the normal movement of the intestines and maintaining a stable internal environment. The identification of commensal bacteria by Toll-like receptors (TLRs) is vital for driving the development and physiological renewal of epithelial cells, thereby safeguarding the epithelial surface against intestinal damage [23,24]. Paneth cells directly detect enteric bacteria via cell-autonomous MyD88-dependent Toll-like receptor (TLR) activation, initiating the production of several antimicrobial factors. Paneth cells were crucial for regulating the barrier’s permeability to both commensal and pathogenic microbes [16,24]. In order to prevent harmful bacteria from multiplying and colonizing the host, the microbiota plays a role in the development of GALT and the immune system as a whole by promoting IgA secretion and the synthesis of antimicrobial compounds [16,25]. The gut microbiota affects the maturation and operation of the mucosal immune system by the use of ligands generated by commensal bacteria, like LPS. By recognizing the transient receptor potential (TLR) of compounds known as pathogen-associated molecular patterns (PAMPs), the innate immune system can identify potentially harmful bacteria. In response, it increases cytokine levels and enhances T-cell activation to combat these pathogens [26]. 

## 4. The Correlation between Endurance Exercise and Intestinal Microbiota

Over the preceding decade, there have been numerous attempts to decipher the connection between exercise and the microbiota in the gastrointestinal tract. Physical activity proposes an assortment of advantageous consequences for human well-being [27]. Body fat, age, nutrition, timing, and training state are among the variables that influence how humans’ gut microbiota compositions are affected by excessive exercise. The benefits of physical activity begin at a young age [28]. Elite athletes tend to have higher gut microbial diversity and shifts towards beneficial bacterial species. The appropriate balance involving level of training, effectiveness, microbiome composition, and activities ought to be assessed to enhance effectiveness and well-being in elite athletes. Moderate and vigorous physical activity are essential elements of endurance athletes’ training programs, each yielding unique health benefits [29]. Endurance activities, like running, skiing, cycling, aerobics, and swimming, improve cardiovascular fitness. As humans engage in exercise at increasing intensities, oxygen consumption rises linearly to satisfy the requirements of the active skeletal muscles, ultimately attaining peak oxygen uptake [30]. A consistent exercise regimen is significantly advantageous for the prevention and management of numerous chronic diseases, especially those impacting the cardiovascular system. Nonetheless, extended and excessive endurance exercise may result in anatomical alterations in the heart and principal arteries. In experienced endurance athletes, recurrent myocardial stress and repair may lead to myocardial fibrosis, especially in the atria, interventricular septum, and right ventricle, thus heightening the risk of atrial and ventricular arrhythmias. Moreover, excessive high-intensity endurance training has been linked to diastolic dysfunction, arterial stiffness, and coronary artery calcification [31]. Numerous pieces of scientific evidence imply the existence of an extensive spectrum of beneficial microbiota in an athlete, including an enormous prevalence of *Bacteroidetes, Akkermansia*, *Veillonellaceae*, *Prevotella*, and *Methanobevibacter* [32]. The presence of these beneficial microbes facilitates energy production, mitigates inflammation, and enhances digestion, all of which are essential for endurance. Consequently, their presence enhances athletes’ endurance and overall performance [32]. Moreover, the relationship between gut microbiota and exercise is reciprocal. Exercise has demonstrated a beneficial effect on the gut mucus layer, which serves as a crucial foundation for mucosa-associated bacteria, such as *Akkermansia muciniphila. Roseburia hominis* and *Faecalibacterium prausnitzii* are capable of producing butyrate, which has been found to have a beneficial effect on intestinal function and lipid metabolism. Additionally, butyrate has anti-inflammatory characteristics [33]. There is an intriguing hypothesis that proposes complex carbohydrates are digested and fermented in the colon, leading to the generation of SCFAs, such as n-butyrate, acetate, and propionate. Propionate and acetate are transported through the circulatory system to various organs, where they serve as substrates for energy metabolism [34]. Therefore, the production of SCFA by the microbes in the gut will enhance the host’s health during exercise, thereby playing a role in the body’s adaptation to exercise. Increased physical activity levels affect host utilization of energy, with some of these effects potentially transmitted through the intestinal flora. Lipids, carbohydrates, and protein are the three primary macronutrients that function as sources of energy in human nourishment; they vary significantly in digestion and hence offer unique microbiota-accessible nutrients [35]. Nevertheless, gut microbiota is essential for optimizing energy extraction from nutrients. These microorganisms decompose complex carbohydrates, fibers, and other substances that the human digestive system is incapable of adequately processing [36]. In this manner, they generate short-chain fatty acids (such as butyrate), vitamins, and other metabolites that furnish supplementary energy and facilitate essential tasks, including immunological regulation and inflammation management [37]. This symbiotic interaction may augment endurance by enhancing nutrition absorption and energy availability. The SCFAs generated during the fermentation of microbiota serve as an energy substrate for the liver and muscle cells, consequently enhancing endurance performance. The overall mechanism is shown in Figure 1. Furthermore, it is necessary to maintain a harmonious equilibrium in the composition of the gut microbiota over a period of time [38].

A reduced diversity of microbes has been associated with various gastrointestinal (GIT) disorders, including Crohn’s disease, small bowel cancers, and diabetes [39]. The microbiota of the human gut, with its capacity to acquire energy, regulates the body’s immune system and affects the health of the gastrointestinal tract, possibly exerting a significant effect on athletes’ overall well-being and performance during sports. Therefore, comprehending the ways by which the microbiota may influence athletic performance is of significant interest to athletes seeking to enhance their competitive performances and minimize recuperation time during training [40].

According to the available scientific discoveries, *Veillonella* is a microorganism that has been identified as having performance-enhancing properties. This microbe is recognized in general for its ability to metabolize lactate and therefore generate propionate [41]. Consequently, the synthesis of SCFAs by the gastrointestinal microbiota is expected to have advantageous effects on the overall well-being of the host organism during physical exertion, consequently playing a role in facilitating adaptations generated by exercise. The SCFAs generated by the process of microbial fermentation subsequently serve as a vital energy substrate for both hepatic and muscular cells, hence enhancing overall endurance capacity [22]. In the context of endurance exercise, the primary constraint is energy availability, and it is imperative to restore cellular energy homeostasis. The interaction between gut bacteria and the host’s energy metabolism is multifaceted [42]. The gut microbiota enhances the capacity to extract energy from ingested food, resulting in the production of metabolites and microbial products such as SCFAs, secondary bile acids, and lipopolysaccharides. The subsequent modulation of hunger, energy absorption and storage, gastrointestinal motility, and energy expenditure will be facilitated by these microbial compounds [35]. The overall association between the gut microbiota and physical activity in athletes [29,43] is described below in Figure 2.

## 5. Involvement of Gut–Brain Axis in Overall Athletic Performance

The gut–brain axis, defined as a sophisticated two-way communication channel interconnecting the gastrointestinal microbiota and the brain’s central nervous system (CNS), possesses a crucial impact on sports persistence and overall athletic performance [44]. The gut–brain axis has a significant impact on sports performance by modulating mood, mental wellness, and cognitive performance by means of the generation of neurotransmitters and the regulation of inflammatory processes. The gut microbiota has the potential to enhance cognitive performance and decrease the activity of the hypothalamic–pituitary–adrenal (HPA) axis [45]. The microbiota in the gut generates an array of compounds called metabolites, which include SCFAs and neurotransmitters like serotonin and dopamine that can influence brain function and mood [46]. Metabolites and neurotransmitters are transported to the brain through the bloodstream and vagus nerve [47]. These metabolites, e.g., SCFAs and neurotransmitters, influence cognitive functions, stress response, and perception of exhaustion. SCFAs and neurotransmitters such as serotonin promote cerebral wellness through mitigating inflammatory response and preserving the blood–brain barrier. Furthermore, they strengthen memory, learning, and focus, hence positively influencing total cognitive performance. Serotonin, frequently referred to as the “feel-good” neurotransmitter, also contributes to mood control, hence affecting cognitive clarity [48]. Stimulating the vagus nerve enhances the strength and stability of the intestinal barrier, decreases inflammation in the surrounding tissues, and suppresses the production of inflammatory signaling molecules [49], which are important elements in athletic performance. Furthermore, the gut microbiota’s ability to regulate systemic inflammation and redox levels can help in reducing muscular pain and enhancing recovery periods, allowing athletes to engage in more efficient training [32]. An optimal gut flora can contribute to the regulation of cortisol, resulting in decreased levels of stress and anxiety [12]. Consequently, this can enhance concentration and stamina during physical exertion, as described in Figure 3.

## 6. Bidirectional Communication between Intestinal Microbiota and Muscle (Gut–Muscle Axis) in Athletic Performance

The microbiota of the gastrointestinal tract has a significant impact on muscle functioning, influencing metabolic processes and overall health, which then impacts sports performance. The gut microbiota plays an integral part in the breakdown and assimilation of vital nutrients required for maintaining muscle strength, including amino acids, vitamins, and minerals. In addition, they generate SCFAs, such as butyrate, propionate, and acetate, which play a crucial role in controlling energy metabolism and decreasing muscle inflammation [50]. SCFAs act as fuel for muscle cells and possess anti-inflammatory characteristics, which aid in enhancing muscle repair and diminishing tiredness [51].

The gut–muscle axis is potentially bidirectional, with the microbiota exerting influence on muscle function, while physical exercise has a role in shaping the composition of the microbiota. The intensity and frequency of exercise play a crucial role in defining the dominant direction of the axis and its physiological and pathological repercussions [52]. Long-term exercise of low-to-moderate intensity positively impacts the microbes in the gut through boosting microbial diversity, increasing beneficial microbe abundance, stimulating SCFA synthesis, and augmenting gut barrier function and immune modulation [53]. Acute moderate-intensity physical activity positively influences the intestinal microbiota of athletes, resulting in enhanced microbial diversity, an increase in beneficial microbes such as Bifidobacteria and *Lactobacilli,* a reduction in potentially pathogenic bacteria, and the promotion of SCFA production, thereby improving gut barrier function and exerting anti-inflammatory effects [54]. Following workouts with high intensity, the energy that the body produces is mostly directed to preserve muscular activity and other physiological functions, thereby limiting the energy available for gut bacteria, thus impacting bacterial growth and processes of metabolism [22]. Metabolites originating from intestinal microbes are complex signaling compounds that play a crucial role in the functioning of muscles. They can affect pathways that are responsible for the loss of skeletal muscle, making them a potential target for further therapy in muscular dystrophy [55]. Skeletal muscle functions as a hormone-secreting gland by releasing growth factors and cytokines, which have systemic effects. Skeletal muscle possesses both metabolic and endocrine characteristics, enabling it to interact with other systems, including the digestive system and the microbial community in the gut [52]. 

One plausible mechanism involves the phenomenon of mitochondrial crosstalk, wherein muscle mitochondria stimulate innate immune responses or impact the functionality of intestinal effector cells (such as immune cells, epithelial cells, and enterochromaffin cells) through the generation of reactive oxygen species (ROS) and reactive nitrogen species (RNS). This subsequently leads to modifications in signaling processes within the digestive tract [42]. A summary of the way muscles can affect gut microbial populations is given in Figure 4. Contracting skeletal muscle also has the capacity to produce myokines, cytokines, and proteins that elicit autocrine, paracrine, or endocrine effects [56]. The most extensively researched myokine is interleukin (IL)-6. An increase in systemic IL-6 levels can impact the gastrointestinal milieu through the activation of intestinal L-cells, resulting in the release of glucagon-like peptide 1 (GLP-1). GLP-1, an incretin hormone, exerts its effects on the β-cells of the pancreas by augmenting insulin secretion. Additionally, it serves to reduce intestinal motility and increase satiety, facilitating the availability of nutrients [57]. Moreover, the association between the composition of the gut microbiota and an individual’s condition extends beyond physical activity. Individuals who suffer from age-related sarcopenia exhibit distinct gut microbial profiles characterized by a reduction in bacteria that produce SCFAs. The alteration of gut microbial status through endocrine signaling can be influenced by exercise and muscular use [55,58,59]. The involvement of the gut–muscle axis in athletes’ endurance is depicted below in Figure 4.

## 7. Optimizing Athletic Performance with Microbiome Intervention Strategies

Microbiome research has focused extensively on lifestyle variables such as physical activity along with other intervention tactics [60]. In response to many different perturbations in the gut microbiome, studies have investigated the correlation between host health and characteristics of the gut microbiota, such as diversity and the abundance or lack thereof of particular taxa [61]. Personalized nutrition plans that consider each athlete’s unique needs, physiological demands, and health and fitness objectives are quickly replacing the generic “one size fits all” approach [62]. Multiple research studies have identified a significant association between interventions aimed at manipulating the gut microbiota and the physical performance of individuals engaged in sports activities. Intervention strategies such as incorporating probiotics, prebiotics, synbiotics, and postbiotics, as well as modifying dietary choices, have shown promise in enhancing athletic performance through various mechanisms [43]. Probiotics have the ability to enhance the function of the intestinal barrier, leading to a reduction in systemic inflammation and an improvement in immunological function. These effects are essential for maintaining optimal health and performance [63]. Prebiotics function as nourishment for advantageous intestinal bacteria, stimulating their proliferation and functionality, resulting in enhanced assimilation of nutrients and energy utilization [64]. Modifying one’s dietary habits to include more foods that are high in fiber can have a beneficial impact on the composition of the gut microbiota. This can lead to an increase in the growth of beneficial microbial species that produce SCFAs. These SCFAs play a critical role in energy generation and muscle function, directly affecting athletic performance. In addition, fecal microbiota transplantation, albeit in the experimental stage, has demonstrated promise in restoring a harmonious balance of gut microbiota, hence improving overall health and performance outcomes [65]. These types of interventions provide an encouraging effect on gut health and also influence the gut–brain–muscle axis, contributing to enhanced mood, less fatigue, and faster recovery. Consequently, athletes that utilize gut microbiota therapies may encounter heightened endurance, strength, and overall performance. The increasing comprehension of the gut microbiota’s impact on athletic performance opens up possibilities for customized nutrition and training approaches, designed to enhance an individual’s microbiome for optimal athletic success [66]. Considering the scientific evaluations from both Scheiman et al. [41] and Lee et al. [67,68] utilized intervention strategies that could directly or indirectly improve the production and/or availability of SCFAs, it is feasible that these measures would have contributed to a rise in SCFA availability. The research conducted by Scheiman et al. [41] indicated that *Veillonella* was markedly elevated in the gut microbiota of marathon runners following strenuous activity. This bacterium transforms lactic acid, generated during exercise, into propionate. Propionate promotes athletic performance by serving as an additional energy source and mitigating muscular exhaustion. *Veillonella* enhances the availability of SCFAs, particularly propionate, which aids in improving athletic endurance by supplying a rapid energy source during vigorous activity. This, in turn, could have resulted in one or more of the following effects: improved storage of glycogen, increased availability of glucose, enhanced oxidation of fatty acids, and preservation of endogenous glucose. Therefore, any single element or a combination of these factors could have increased the capacity for endurance exercise. Lee et al. (2019) [67] offer significant insights into the impact of various probiotic strains, notably *Bifidobacterium longum* OLP-01, on athletic performance. The principal findings indicate that supplementation with *Bifidobacterium longum* OLP-01 considerably enhances exercise endurance and diminishes tiredness in animal models. The probiotic enhanced glycogen accumulation in muscles and the liver, which is essential for sustained energy provision during vigorous physical exertion. It also facilitated a decrease in indicators of muscle injury and inflammation following exercise, thereby enhancing recovery. This study further shows that *Bifidobacterium longum* OLP-01 might be useful for improving gut health and athletic performance by speeding up recovery and improving energy metabolism, which is useful for sports nutrition and performance improvement. The study conducted by Lee et al. (2020) [68] reveals that *Lactobacillus salivarius* SA-03 has significant ergogenic effects, akin to those of the previously examined *Bifidobacterium longum* OLP-01. This specific strain of bacteria not only augments endurance performance but also mitigates exercise-induced tiredness and promotes recovery. The outcome of this study indicates that *Lactobacillus salivarius* SA-03 may serve as a useful probiotic for athletes and individuals aiming to enhance exercise performance and mitigate post-exercise exhaustion, thus constituting a significant enhancement to sports nutrition techniques.

## 8. Diet as One of the Influencers in Athletic Endurance

Improving performance during exercise can be achieved by modulating the microbiome of the intestinal tract by means of dietary strategies, considering nutritional intake possesses an immediate impact on microbial communities and metabolites. The gastrointestinal microbes’ biology and behavior are impacted by food type, quality, and origin, which in turn affect host–microbe interactions [69]. The domain of sports performance incorporates a dynamic interaction of multiple factors, with dietary practices and body composition serving as crucial elements [70]. In general, the athletes consumed more high-calorie food, fat, carbohydrates, sweets, protein, protein supplements, and saturated fat on a daily basis. Hence, deriving definitive conclusions regarding the influence of nutrition on gut microbial diversity and exercise performance is challenging based on the study conducted by Clarke et al. [71,72]. As we explore the scientific foundations of the correlation between nutrition, gut flora, and athletic endurance, it becomes increasingly apparent that achieving peak athletic performance extends beyond the realm of conventional physical training [43]. A comprehensive dietary regimen encompasses the essential macronutrients, namely carbohydrates, lipids, and proteins, as well as micronutrients such as vitamins and minerals, which are vital for athletes in terms of fuel generation and optimal nutrient assimilation during physical activity [73]. Carbohydrates, such as those found in food, serve as a fundamental and principal energy source for individuals engaged in endurance-based athletic activities [74].

Endurance athletes encounter challenging circumstances during their physical activity regimens, prompting them to explore alternate dietary approaches aimed at enhancing both their endurance performance and metabolic well-being [75]. Macronutrients, which include carbohydrates, proteins, and fats, play a critical role in athletic nutrition. Adequate carbohydrate intake is necessary to support energy production and replenish glycogen stores, thereby reducing the likelihood of restrictive eating behaviors [76]. Meanwhile, healthy fats contribute to sustained energy, hormone production, and overall health, promoting a balanced approach to nutrition [77]. Besides macronutrients, micronutrients, which include vitamins and minerals, are essential for energy metabolism and immunological function. Advocating for a varied, nutrient-rich diet is essential for ensuring athletes have sufficient micronutrients, therefore mitigating the risk of nutritional deficits that could lead to disordered eating [78]. Proteins are indispensable for muscle repair and growth, emphasizing the importance of meeting increased protein needs without resorting to excessive dietary restrictions [79]. Moreover, the influence of protein source, encompassing its quality and digestibility, on the location of fermentation within the gastrointestinal tract may be significant. Proteins with high digestibility, such as whey protein, have the ability to be broken down by the enzymes present in the proximal intestine of the host organism [80]. This process leads to a decrease in microbial fermentation. In a similar vein, proteins derived from plants can undergo microbial fermentation in a more distant location due to incomplete digestion by enzymes within the host, especially when the protein concentration is elevated [81]. The available evidence suggests that proteins derived from vegetables have a more pronounced impact on microbial diversity compared to animal proteins; however, further research is required to investigate this phenomenon specifically in athletes [82]. The Mediterranean diet is often regarded as a model of well-balanced nutrition, as it enhances endurance and promotes the development of diverse and thriving gut bacteria [83]. The overall mechanism of diet influencing the athlete’s endurance is shown in Figure 5. 

## 9. Prebiotics, Probiotics, Synbiotics, and Postbiotics in Athletic Endurance

The gut microbiota and athletic performance are essential; we must explore ways to leverage this knowledge to enhance current sports nutrition and positively regulate the gut flora. A strategy for this is to employ microbiome-modifying agents such as probiotics, prebiotics, and synbiotics [15]. A prebiotic is a substance that is specifically used by microbes in the body, which provides a health advantage. It is present in non-digestible oligosaccharides, fructans, and galactans [84]. Postbiotics, a relatively new term, refers to the use of non-living microbes or their components to provide health benefits to the host. The International Scientific Association of Probiotics (ISAPP) typically defines postbiotics as “creating with inactive microorganisms and/or their components that convey a physiological impact” [85] in the host. Postbiotics, on the contrary, extend much beyond the limited definition of a “dead probiotic”; they can include whole or fragmented cells, as well as large or little pieces of the originating bacterium. Exopolysaccharides, peptidoglycan, lipoteichoic acid, short-chain fatty acids, amino acids, and probiotics are all examples of metabolites that can be produced by both active and inactive bacterial cells [86]. These metabolites possess the capacity to regulate the gut microbiome and improve multiple facets of health [87]. Cheng et al. [88] performed a study that used a heat-killed variant of *L. plantarum* TWK10 to investigate the effects of six weeks of supplementation in 30 healthy males with either a placebo or 3 × 10^10^ heat-killed cells of *L. plantarum* TWK10. The research did not include any kind of exercise program; rather, participants were given supplements and then tested before and after the supplementation period to see how their endurance exercise performance, muscle mass, body composition, stress levels, and fatigue levels changed. When compared to the placebo group, those who took the postbiotic supplement showed improvements in muscular mass, grip strength, and endurance performance. The probiotics now authorized for human use include *Bifidobacterium*, *Lactobacillus*, *Enterococcus*, *Escherichia coli*, and *Bacillus subtilis*, among other strains [89]. Guar gum, inulin, lactulose, polydextrose, goat milk oligosaccharides, and gum-arabic are among the most prevalent prebiotics. Prebiotics that have not been digested are taken to the large intestine to be broken down and used by the good bacteria that live there. Intestinal epithelium absorption or portal vein transfer to the liver of the generated secondary metabolites eventually impacts the body’s physiological functions [90]. An extensive amount of recent research has shown that plant-based polyphenols and flavonoids can modulate the overall composition of gut microbes and the function of the intestinal barrier, hence increasing the host’s energy level [91,92]. 

Extended and strenuous training can result in respiratory infection, gastrointestinal problems, inflammatory disorders, immunosuppression, oxidative stress, anxiety, and exhaustion. The rigorous physical activity stimulates heightened permeability of the intestinal mucosa and elicits inflammatory reactions [93]. Various scientific discoveries indicate that probiotics have beneficial effects on athletes, particularly those belonging to the *Lactobacillus* and *Bifidobacterium* species [94]. Numerous probiotics, particularly multispecies formulations, are believed to effectively reduce gastrointestinal symptoms and upper respiratory symptoms, and may also enhance post-exercise recovery. The post-recovery effects are contingent upon the species, dosage, duration, and method of administration (e.g., capsules, sachets, fermented milk). The molecular processes underlying the efficacy of probiotics in the context of sports remain unexplained. *Lactobacillus* species can impact IgA secretion and release through the pathway involving IFN-g-generating Th1 cells [95]. These kinds of microbes stimulate the immune system in the gastrointestinal tract through interactions with intestinal epithelial cells, M cells, and dendritic cells. The gastrointestinal epithelium communicates with the upper respiratory tracts, potentially elucidating the enhancement in gastrointestinal symptoms and the severity of upper respiratory symptoms [96]. Some strains of probiotics activate TLR2 signaling via their molecular patterns (PAMPs), resulting in an inflammatory response through the NF-kB pathway. The inflammatory mediators may induce beneficial modifications in the intestinal barrier to regulate this response [94].

Synbiotics refers to an amalgamation of probiotics and prebiotics, which function synergistically to boost the lifespan of probiotic microorganisms in the gastrointestinal system [97]. In a prior research investigation conducted by West et al. [98], a double-blind controlled trial was carried out over a period of 21 days to investigate the effects of synbiotic supplementation. The supplementation included specific strains of bacteria (*Lactobacillus paracasei 431*, *Bifidobacterium animalis* ssp. *lactis BB-12*, *L. acidophilus LA-5*, *L. rhamnosus LGG*), as well as raftiline, raftilose, lactoferrin, immunoglobulins, and acacia gum. The results showed that the synbiotic supplementation led to a smaller increase in serum IL-16 concentration compared to the group that received acacia gum. However, there were no significant differences observed in fecal SCFA concentrations, mucosal immunity, or gastrointestinal tract permeability between the two groups. In addition, a study conducted by Quero et al. [99] implemented a triple-blinded design and found that professional soccer players who were given synbiotics (specifically *Bifidobacterium lactis* CBP001010, *Lactobacillus rhamnosus* CNCM I-4036, *Bifidobacterium longum* ES1, and fructo-oligosaccharides) experienced significant improvements in anxiety, stress, and sleep quality.

These findings indicate that the varying impacts of synbiotics may be associated with the frequency and intensity of physical activity, as well as the length of the study. Synbiotics have the potential to provide several health advantages to the host, including the regulation of gut microbiota, relief from gastrointestinal symptoms, enhancement of immunity, reduction in inflammation and oxidative stress, and improvement of blood lipids.

## 10. Potential Mode of Action of Probiotics on an Athlete’s Endurance

Research has demonstrated that taking probiotics can improve health, particularly by calming the immune system and strengthening the gut mucosal barrier. There are many defense systems that support the wall of the intestinal tract [100]. These include the mucous membrane, antimicrobial peptides, secretory IgA, and the epithelial junction adhesion complex. When the barrier function is not working properly, bacteria and food antigens can get into the submucosa and cause inflammatory responses [101]. Anderson et al. [102] stated that upregulating genes involved in tight junction signaling could be one way to improve the intestinal barrier’s ability to keep bacteria out. Probiotics may improve barrier function and infection prevention by encouraging mucus production, among other possible mechanisms [63], as shown in Figure 6. Researchers have discovered that several strains of *Lactobacillus* can enhance the expression of mucin in human intestinal cell lines. Therefore, these species may help restore the mucus layer in cases where the mucosa is compromised [103]. 

## 11. Clinical Studies Performed on Athletes for Better Endurance Using Probiotics

Athletes who are thinking about using these techniques to boost their performance should consult with healthcare providers or sports nutritionists beforehand. Expertly shaped to meet the specific needs of each athlete, these specialists may provide personalized advice based on the most recent scientific research. In addition, more clinical trials are expected to provide more information and recommendations about the use of probiotics, prebiotics, and postbiotics in sports performance. The predominant probiotic microorganisms commonly utilized are the different strains of *Lactobacillus* or *Bifidobacterium* [19,20,104,105,106]. However, other bacterial and yeast strains, such as *Escherichia coli* and *Saccharomyces*, are also frequently utilized. The evaluated bacterium strains encompassed species such as *Bacillus subtillis* [107], *Bacillus coagulans* [108], *Veillonella atypica* [41], and *Saccharomyces boulardii* [109]. Multiple studies have shown evidence that the makeup of the gut microbiota and efforts to modify it can have a positive effect on endurance. A highly significant study in this field found that *Veillonella atypica*, a bacterial species known for its capacity to metabolize lactate, was more abundant in the gastrointestinal microbiomes of individuals who took part in marathon races, as reported by Scheiman et al. (2019) [41]. The researchers hypothesized that the lactate metabolism of *Veillonella atypica* was increasing the rate at which lactate is processed, leading to a decrease in the build-up of lactic acid in muscles and ultimately improving endurance. Shing et al. (2014) [106] conducted a randomized, double-blind, placebo-controlled trial including endurance athletes, as documented in an article that appeared in the International Journal of Sports Physiology and Performance. The individuals who received a probiotic regimen comprising *Lactobacillus acidophilus* and *Bifidobacterium lactis* for a duration of 12 weeks exhibited a noteworthy decrease in gastrointestinal discomfort while engaging in physical training and competitive activities. The observed decrease in gastrointestinal problems was found to be linked with enhanced endurance performance, as athletes reported experiencing reduced discomfort and were able to maintain higher levels of exertion for extended periods of time. A research investigation employing a double-blind, randomized, placebo-controlled, crossover design was carried out with a 21-day washout period. A total of 15 participants who were in good physical condition ingested an encapsulated probiotic containing various strains of *Bifidobacterium* (*B.*) *breve* BR03 as well as *Streptococcus* (*S.*) *thermophilus* FP4. It was administered with a concentration of 5 billion live cells (AFU) each, or a placebo, and the duration was every day for a period of 3 weeks before participating in exercise that induced muscle damage. The results of the investigation indicated that the cohort of athletes who consumed the probiotic supplement had decreased levels of systemic inflammation and indications of oxidative stress during endurance exercise, as compared to the control group receiving a placebo. Furthermore, the experimental group that received probiotics had a shortened time to exhaustion during endurance trials. This suggests that the observed reduction in inflammation could potentially contribute to enhancing endurance capacity [110]. Furthermore, the potential of *Lactobacillus plantarum* PS128 to reduce inflammation and oxidative stress while simultaneously enhancing exercise performance was explored in an intervention study by Huang et al. in 2020 [111]. Additional research looked into the microbiome, blood cells, biochemistry, body composition, and associated metabolites in addition to endurance performance and body composition. PS128 considerably outperformed the placebo group in terms of endurance, increasing it by around 130% [111].

A challenging scientific investigation, executed by the research team of Robert et al. [112], aimed to assess the potential effects of a combination of different strains of probiotics and prebiotics, in conjunction with or without antioxidant supplementation, on the levels of endotoxin and intestinal integrity in triathletes. In this experiment, for a twelve-week intervention period, far-flung athletes competing in triathlon were given a probiotic blend mixture comprising two particular species of *Lactobacillus acidophilus* as well as *Bifidobacterium bifidum*, in conjunction with prebiotics in the form of fructo-oligosaccharides. The overall outcome showed that the cohort of individuals who were administered probiotics, prebiotics, and antioxidants as supplements had a significant reduction in endotoxin components across both pre-race and post-race evaluations. The research conducted in this study revealed a noteworthy rise in gastrointestinal permeability, as seen by the urine lactulose/mannitol recuperation percentage, within the placebo group when compared to the initial measurements. In contrast, a statistically insignificant rise in the permeability of the intestinal tract was observed in both groups receiving probiotics when compared to their respective baseline measurements. The individuals from the probiotic groups exhibited a decrease in gastrointestinal symptoms in comparison to the placebo group, whereas the treatments did not result in any discernible impact on race performances. The findings collectively indicate that the ingestion of multi-strain probiotic supplements has the potential to enhance the gastrointestinal well-being of long-distance runners. The inclusion of antioxidants in the supplement has the potential to enhance the positive effects by mitigating the levels of endotoxins in athletes throughout the endurance-training phase. Lamprecht et al. (2012) [19] conducted a study that aimed to investigate the effects of a probiotic supplement, specifically comprising a particular strain of *Lactobacillus casei Shirota*, on immune system activity as well as endurance performance among individuals who perform running for long periods of time. Interestingly, the individuals who were administered probiotics demonstrated a reduction in indicators of inflammation and an augmentation in the activity of natural killer cells, indicating an improved immune response. Furthermore, it was observed that athletes belonging to the probiotic intervention group exhibited enhanced running performance during a half-marathon event, suggesting that the consumption of probiotics could potentially enhance endurance capabilities and provide immune system support for runners. Further work described by Liu et al. [113] employed a double-blind experimental design in order to investigate the potential effects of *L. plantarum* PS128 supplementation. Participants were instructed to adhere to their usual daily routine for a period of 24 h before any experimental assessment. They were specifically advised to refrain from indulging in physically demanding activities, staying awake late at night, smoking, and consuming alcoholic beverages. During the administration of the supplements, the participants were instructed to abstain from ingesting probiotics, prebiotics, fermented goods such as yogurt or other meals, vitamins, herbal extracts, and antibiotics. This precautionary measure was implemented to prevent any potential interference that could compromise the integrity of the experimental phases. The probiotic strain *L. plantarum* PS128 has demonstrated favorable impacts on the preservation of exercise performance. It overall modulates inflammation, oxidation, and metabolism. In summary, integrating prebiotics, probiotics, synbiotics, and postbiotics into the nutritional intake of an athlete can yield significant advantages in terms of endurance and overall performance. These components synergistically function to maximize gastrointestinal health, augment food assimilation, diminish inflammation, and bolster a resilient immune system. Consequently, athletes might experience greater digestion, diminished gastrointestinal problems, accelerated recuperation, and heightened energy levels, all of which contribute to superior sports performance.

## 12. Conclusions

The immense relevance of nutrition in the everyday lives of athletes cannot be overemphasized, serving as an essential foundation underlying performance, recovery, and overall well-being. As players consistently push their physical boundaries, the complex connection between dietary habits and athletic results has become a focal point in the field of sports science. Each intervention, including the careful creation of tailored nutrition plans, the use of advanced technologies, and the implementation of psychological strategies, plays a crucial role in the holistic approach to improving athletes’ nutrition. The correlation between physical fitness and the microbiota is multifaceted. The engagement in exercise and the pursuit of physical fitness have been found to have a positive impact on the makeup of the microbiome, leading to various health benefits for the host. As a result, these practices can contribute to the preservation and enhancement of athletic potential. Research indicates that various types of physical activity have the potential to impact the prevalence of distinct bacterial communities, consequently leading to alterations in the composition of the microbiome. In brief, the utilization of advantageous microorganisms, such as probiotics, has the potential to foster well-being among athletes and augment their physical prowess and capacity for activity. In addition, it is important to conduct rigorous clinical trials with a sufficient sample size in order to elucidate the specific contributions of gut microbiota populations and probiotics to physical performance, as well as the underlying mechanisms responsible for their possible advantages. Subsequent investigations should strive to elucidate the most effective dosage for each distinct strain that has demonstrated therapeutic potential. Additionally, it is imperative to ascertain if the probiotic strains under scrutiny develop a permanent presence within the intestinal tract or merely exist as transient bacteria that confer advantageous outcomes. 

The influence of prebiotics, probiotics, and postbiotics on the population of athletes symbolizes a burgeoning field of study that is characterized by a relatively modest body of research conducted thus far. There is a prevailing consensus that sportsmen commonly experience a range of symptoms, including respiratory, allergy, and gastrointestinal manifestations. In the context of these symptoms, it has been observed that food and nutrition can potentially provide a protective effect. Consequently, there has been a growing trend in the utilization of therapeutic dietary interventions. It highlights a range of developing findings pertaining to the activities of probiotics in enhancing immune function and mitigating psychological stress through the gut–brain axis (GBA). However, additional tactics must be implemented to attain validated findings. Future research endeavors should prioritize the integration of methodological approaches and refrain from engaging in biased interpretations of findings. Additionally, it is imperative to assess the physiological and clinical significance of the in vitro and ex vivo impacts of probiotic, prebiotic, and/or postbiotic products. This evaluation is particularly crucial in the context of recommending these potential nutritional interventions to athletes.

The future trajectory of sports nutrition investigation is incrementally converging into the premise of customized nutrition, emphasizing the distinct interaction between nourishment and athletic performance. This evolution reflects the heterogeneity between people’s responsiveness to nutritional treatments, which are impacted by multiple variables like inheritance, microbial community setup, and metabolic rate. The emerging subject of nutrigenomics, which investigates the interplay between nutrition and genes, is poised to have a crucial impact. Researchers can boost overall athletic performance by developing more sophisticated nutritional strategies that optimize body composition and improve recovery through understanding genetic variants that impact nutrient metabolism and dietary responses.

## Figures and Tables

**Figure 1 ijms-25-10076-f001:**
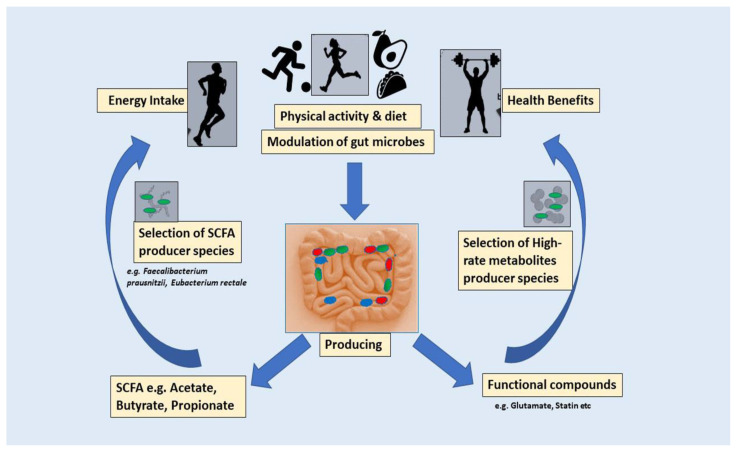
The impact of physical activity on the gut microbiota and the effect of gut microbiota on human health and athletes’ performance through SCFA fermentation from dietary fiber.

**Figure 2 ijms-25-10076-f002:**
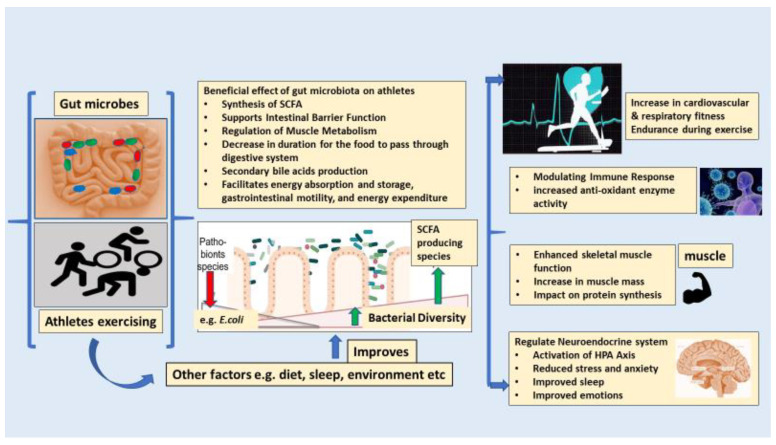
The positive impacts of physical activity and changes in the gut microbiota. Physical activity triggers advantageous molecular changes that promote improvements in cardiovascular and respiratory fitness. The bacterial diversity is enhanced, with a rise in species that produce SCFAs. In contrast, pathobionts such as *E. coli* or *E. faecalis*, which are potentially disease-causing organisms, are normally found as harmless symbionts. The presence of SCFA-producing microbes inhibits potential pathobionts by outcompeting them for resources and sustaining a balanced gut environment due to physical activity.

**Figure 3 ijms-25-10076-f003:**
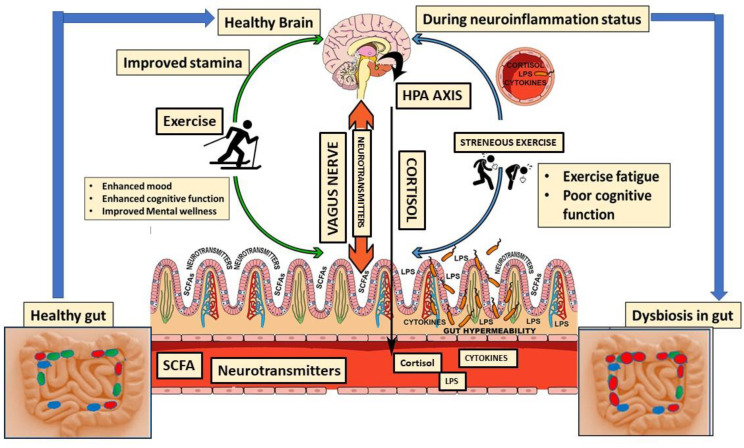
**The figure** demonstrates the mechanism of the HPA axis in athletic endurance.

**Figure 4 ijms-25-10076-f004:**
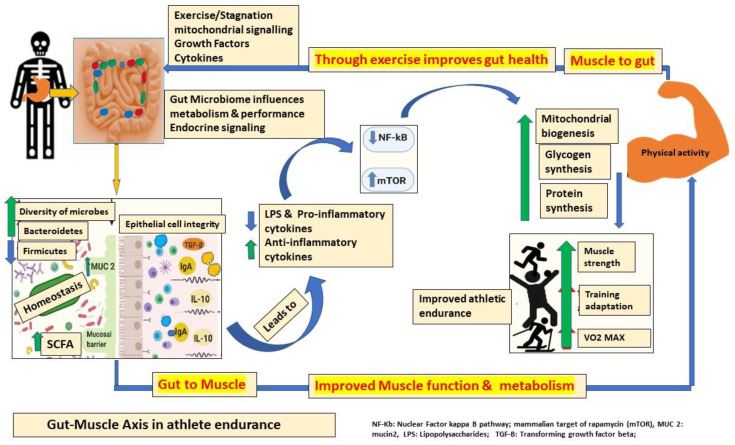
The bidirectional gut–muscle axis denotes the reciprocal communication between the gastrointestinal tract and the muscular system. The figure provides an overview of the main factors that come from both muscle and microbiota, as well as the pathways via which they communicate and the effects they have on muscle. Engaging in physical activity and using our muscles can alter the makeup of gut microbes by transmitting hormonal signals. The gastrointestinal tract possesses the capacity to produce metabolites that exert a favorable influence on the development of skeletal muscles. Physical activity and muscle utilization can modify gut microbial composition through endocrine communication. The gut can subsequently generate chemicals that positively affect skeletal muscle development.

**Figure 5 ijms-25-10076-f005:**
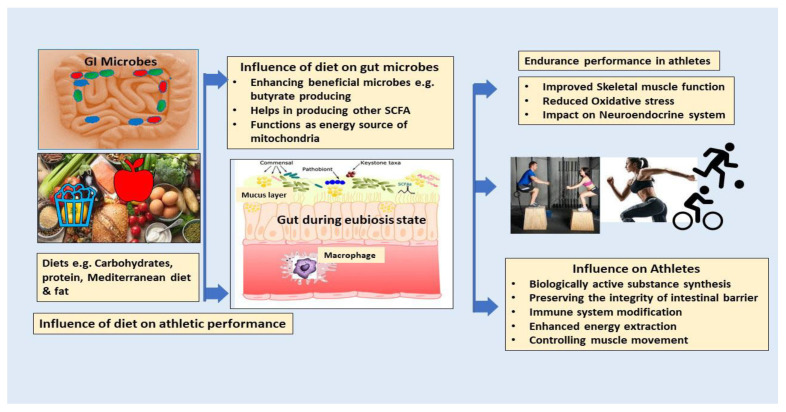
Diet influences athletic endurance through improved muscle function, boosting the immune system, and supporting gut barrier integrity.

**Figure 6 ijms-25-10076-f006:**
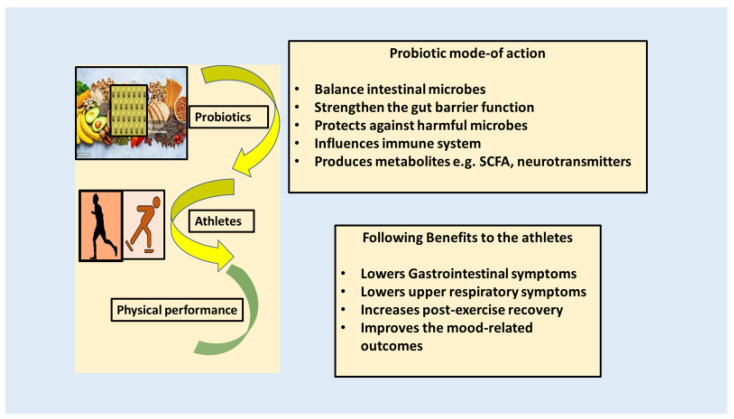
The impact of probiotics on the endurance performance of athletes. Research has revealed that probiotics can have an indirect impact on sports performance by enhancing many factors such as the immune system and response to upper respiratory tract infections (URTIs), reducing oxidative stress, and better monitoring planned exercise sessions.

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
