# Peer review of "Intestinal Microbiota Interventions to Enhance Athletic Performance—A Review"

_ijms, 2024, doi:10.3390/ijms251810076_

Round 1

Reviewer 1 Report

Comments and Suggestions for Authors

This review article covers an interesting area and is well written. I only have a few comments below.

Line 58-59: Additionally, there are around 2000 uncultured species in human GI tract.

https://www.nature.com/articles/s41586-019-0965-1

Section 3: regarding SCFA as an energy substrate for the body. The energy from SCFA is essentially from food, and the bacteria have utilized a portion of the energy for its own growth. In terms of host endurance, would it not be more energy efficient to not have the bacteria process the food and utilized by the host directly?

In Figure 4, there is primary one direction from microbiome to the muscle, not too much illustration on the muscle to microbiome direction.

Section 7: Diet seems to affect host endurance directly and not through microbiome. I would recommend reducing the content in this section to focus on the main point of the review.

Section 9: Please change subtitle to “potential” mode of action on an Athlete's Endurance.

Content on section 8 and 10 should be rearranged. I would suggest having a separate section for probiotics only, and another for prebiotics, synbiotics, etc.

All the references are not readable due to lack of space in between words.

Reviewer 2 Report

Comments and Suggestions for Authors

Please see attached 

Comments on the Quality of English Language

English language needs moderate editing.

Round 2

Reviewer 1 Report

Comments and Suggestions for Authors

The manuscript is ready for acceptance.

Author Response

Thank you very much for your valuable comments and suggestions.

Reviewer 2 Report

Comments and Suggestions for Authors

Comments on the Quality of English Language

Minor English language changes required

Author Response

Please see attached file 16092024 Response to reviewer 2
